# Identification of Reactive Oxygen Species and Mechanism on Visible Light-Induced Photosensitized Degradation of Oxytetracycline

**DOI:** 10.3390/ijerph192315550

**Published:** 2022-11-23

**Authors:** Yibo Zhang, Qian Chen, Hao Qin, Junhan Huang, Yue Yu

**Affiliations:** 1School of Emergency Management, Xihua University, Chengdu 610039, China; 2Sichuan Environment and Engineering Appraisal Center, Department of Ecology and Environment of Sichuan Province, Chengdu 610000, China

**Keywords:** oxytetracycline, photolysis, photosensitization, visible light, reactive oxygen species, transformation product

## Abstract

This study investigated the photolysis and TiO_2_-assisted photosensitized degradation of oxytetracycline (OTC) under visible light, the active reactive oxygen species (ROS), and the degradation mechanisms in these two reactions. The results show that the deprotonated OTC could be photolyzed more easily under visible light because of the redshift of its absorption spectrum at high pH values. Due to the TiO_2_-assisted self-photosensitized degradation of OTC, OTC removal in the visible light/TiO_2_ system was more efficient with the addition of TiO_2_, as demonstrated when TiO_2_ was replaced with insulator SiO_2_. The study’s ROS scavenging experiments show that superoxide radical anion (O_2_^•−^) ROS was most responsible for the self-sensitized degradation of OTC in both reactions. OTC degradation under the visible light/TiO_2_ system was enhanced with increasing TiO_2_ load, while the elimination of total organic carbon (TOC) was very limited after 5 h of visible light irradiation. Based on the eight identified transformation products found, five potential reaction mechanisms, including hydroxylation, quinonization, decarbonylation, de-methylation, and dehydration, were proposed for the photolytic and TiO_2_-assisted photosensitized degradation mechanisms of OTC under visible light. This study indicates that OTC can degrade under visible light with or without a semiconductor when conditions are suitable.

## 1. Introduction

Since the late 1990s, pharmaceuticals and personal care products (PPCPs) have been recognized as emerging contaminants (ECs) which are an increasing environmental issue of global concern [1,2]. Oxytetracycline (OTC), a commonly used broad-spectrum antibiotic, has been detected frequently in different aquatic environments worldwide, such as sewage, river water, lake water, seawater and groundwater, due to its extensive use in human and animal disease treatment and livestock growth promotion [3,4,5]. Its occurrence in the environment may induce drug-resistant bacteria development and antibiotic resistance genes (ARGs), which could adversely impact human health and ecosystem stability [6,7,8,9,10]. Therefore, OTC pollution and its removal in contaminated waters have recently received special attention and scientific interest.

Similar to other organic sensitizers, such as dyes, OTC is a confirmed photosensitizer that may undergo self-photosensitization and self-decompose under suitable conditions [11,12,13]. It can be removed via photolysis under (simulated) solar light irradiation in water through direct photolytic degradation and self- photosensitized degradation [14,15,16]. The photolysis of OTC may generate reactive oxygen species (ROS), such as singlet oxygen (^1^O_2_), superoxide radical anions (O_2_^•−^), hydrogen peroxide (H_2_O_2_), and hydroxyl radicals (HO^•^), but their role in the self-photosensitized degradation of OTC is inconsistent in the published literature [17,18,19]. Zhao et al. [12] concluded that under simulated solar light irradiation, the OTC photolytic mechanism comprised direct degradation and ^1^O_2_ oxidation at pH 8.5 and pH 11.0, while only direct degradation was responsible for OTC photolysis at pH 5.5. Chen et al. [17] detected H_2_O_2_ formed through O_2_^•−^ dismutation and ^1^O_2_ in the photolysis of structurally similar tetracycline (TC) under simulated sunlight, although O_2_^•−^ was not observed. However, Niu et al. [20] did not find ^1^O_2_ in irradiated TC under different light wavelengths, but the existence of O_2_^•−^ and its mediated self-sensitized oxidation were verified, indicating reactivity between TC and O_2_^•−^. Therefore, the species of ROS (e.g., ^1^O_2_ and O_2_^•−^) that contribute to OTC or TC self-photosensitized degradation under solar light is still contended. To our knowledge, there are no studies on the identification of ROS in OTC photolysis under visible light and its photolytic mechanism.

Compared to photolysis, OTC degradation efficiency under simulated sunlight irradiation was enhanced in the presence of commercial TiO_2_ (P25) [21,22,23]. A similar phenomenon was also observed in OTC photocatalytic degradation with an NF-TiO_2_ film under visible/solar light. Probable reaction pathways proposed included direct photolysis, UV/vis light-induced photocatalytic oxidation and reduction and visible light-induced self-sensitized oxidation and reduction [12,24]. However, visible light-induced self-photosensitized degradation and its contribution to OTC removal cannot be confirmed because the band gap value of the NF-TiO_2_ used was 2.85 eV, which can be activated under visible light to produce electrons [12]. In OTC photocatalytic degradation with P25 under simulated solar light, TiO_2_ can be excited by the UV spectrum. Therefore, its role in OTC degradation under only visible light cannot be verified. To demonstrate the feasibility of visible light-induced self-photosensitized OTC degradation, semiconductors with no visible light activity and visible light above 400 nm should be used simultaneously. The current literature on OTC photodegradation using TiO_2_ under visible light irradiation is very limited.

The main objective of this work was to investigate the photolysis and TiO_2_-assisted photosensitized degradation of OTC under visible light. The photolytic and photosensitized degradation efficiencies of OTC at different pH values were evaluated. The ROS that most contributed to OTC destruction in these two reactions was also identified. In addition, the effect of TiO_2_ loading on OTC removal and the mineralization of OTC in the visible light/TiO_2_ system were explored. Finally, the photolytic and TiO_2_-assisted photosensitized degradation mechanisms of OTC were studied based on the detected transformation products. This work can provide insightful information on the visible light-induced photolysis and photosensitized OTC degradation through the identification of active ROS and the detailed reaction mechanisms.

## 2. Materials and Methods

### 2.1. Materials

Oxytetracycline hydrochloride (C_22_H_24_N_2_O_9_∙HCl, >95%) was obtained from Aladdin (China) and used without further purification. Commercial TiO_2_ with a BET surface area of 50 m^2^ g^−1^ and an average particle diameter of 30 nm was purchased from Rhawn (Shanghai, China). Silicon dioxide (SiO_2_) nanopowder (particle size 10–20 nm) and catalase were purchased from Rhawn (Shanghai, China). Sodium azide (NaN_3_), *p*-benzoquinone (C_6_H_4_O_2_) and tert-butanol (t-BuOH) were obtained from Kelong (Chengdu, China) and used as received. All aqueous solutions were prepared with ultrapure water (18 MΩ · cm).

### 2.2. Analysis

A high-performance liquid chromatograph (HPLC, Shimadzu LC-2030 Plus, Tokyo, Japan) equipped with a photodiode array detector (PDA) was used to quantify the concentration of OTC. Detailed measurement information was referenced in previous work by Liu et al. (2016) [25]. The transformation byproducts were detected and identified by an ultra-performance liquid chromatograph coupled with a quadrupole time-of-flight mass spectrometer (UPLC-QTOF/MS, Waters Xevo G2-XS QT, Framingham, MA, USA). A sample volume of 5 μL was injected into a C18 column (2.1 × 50 mm, 1.7 μm). Mobile phase A was a 0.1% formic acid water solution, and mobile phase B was a 0.1% formic acid acetonitrile solution. Gradient elution was programmed as follows: 5% B linearly increased to 90% in the initial 5 min, maintained for 1 min, and dropped to 5% B in the next 0.1 min. The flow rate was 0.2 mL min^−1^, and the column temperature was 30 °C. Data were analyzed using Masslynx 4.1 software (Waters). Total organic carbon (TOC) was measured using a VCSH-ASI TOC analyzer (Shimadzu, Tokyo, Japan).

### 2.3. Photochemical Experiments

Irradiation experiments were carried out in a bench-scale photochemical apparatus housing two 15 W fluorescent lamps (Philips, Amsterdam, The Netherlands) mounted with a UV block filter (UV420) for simulating visible light. Light below 420 nm was effectively eliminated by the UV block filter. The average visible light intensity measured 1.3 mW cm^−2^ using a radiant power meter (Lutron Corp., Coopersburg, PA, USA). A typical experiment can be described as follows: 10 mL of reaction solution was added to the reactor (borosilicate glass Petri dish), sealed with a quartz cover (60 mm (ø) × 15 mm (h)) to avoid solution evaporation, and then placed in the apparatus under continuous stirring. The initial concentrations of OTC and TiO_2_ were 10 mg L^−1^ and 0.5 g L^−1^ unless noted otherwise. All the reaction solutions were adjusted to the designated pH using a 50 mM phosphate buffer. For the degradation experiments, the irradiated solution (0.2 mL) was sampled after 0, 1, 2, 3, 4, and 5 h, mixed with 0.2 mL of methanol, and filtered with a 0.2 μm syringeless filter before HPLC analysis. For the transformation byproducts and mechanism study using a UPLC-QTOF/MS, at a given time, a 0.2 mL sample was taken without adding methanol and analyzed immediately after filtration. All experiments were conducted in triplicate at room temperature (25 ± 1 °C) except for the investigation for mechanisms. Error bars in the figures represent the standard mean error.

## 3. Results

### 3.1. Photolysis and TiO_2_-Assisted Photosensitized Degradation of OTC under Visible Light

OTC has four existing forms in the whole pH range due to its pKa values of 3.57, 7.49, and 9.88 [21]. This study chose to investigate photolysis and TiO_2_-assisted photosensitized OTC degradation under visible light irradiation at pH 2.0, 5.7, 8.6, and 11.5. These pH values were chosen because they respectively represent the OTC species H_3_OTC^+^, H_2_OTC, HOTC^−^, and OTC^2^. As shown in Figure 1a, OTC could not be photolyzed under visible light irradiation at pH 2.0 and its degradation rate was enhanced with increasing pH. OTC photolysis under visible light at pH 8.6 and 11.5, i.e., deprotonated OTC, and occurred more easily than that for protonated OTC. This could be caused by redshift in the OTC absorption spectrum with increasing pH [25].

With the addition of TiO_2_, OTC removal increased at four different pH values under the study conditions, as shown in Figure 1b. The TiO_2_ in this study was obtained directly from Evonik Degussa and used as received; therefore, it could not be activated under visible light irradiation. The improvement effect was thus ascribed to self-photosensitized OTC degradation with TiO_2_ assistance. The adsorbed OTC on the surface of TiO_2_ could be excited to a singlet state and readily transferred to a triplet state (OTC*) through intersystem crossing under visible light irradiation. The generated OTC* might inject an electron to the conduction band of TiO_2_ for capture by free molecular oxygen to produce ROS, such as O_2_^•−^, H_2_O_2_ and HO^•^, resulting in OTC degradation. This is presented in Equations (1)–(6) [12]. To confirm the role of TiO_2_ in photosensitized OTC degradation under visible light at pH 5.7 and 8.6, the semiconductor TiO_2_ was replaced with the electric insulator silicon dioxide (SiO_2_). The results in Figure 2 show the removal of OTC in the visible light/SiO_2_ system was significantly inhibited, comparable with OTC photolysis under visible light at both pH values. Because the electrons produced from OTC* could not be transferred to the SiO_2_ insulator, ROS might not be generated in the visible light/SiO_2_ system. This result indicates the importance of TiO_2_ on the degradation of OTC in the visible light/TiO_2_ system.
(1)$$\mathrm{OTC}+hv\;(>420\;\mathrm{nm})\to {\mathrm{OTC}}^{*}$$
(2)$${\mathrm{OTC}}^{*}+{\mathrm{TiO}}_{2}\to {\mathrm{OTC}}^{+•}+{\mathrm{TiO}}_{2}\left(e^{-}\right)$$
(3)$$e^{-}+O_{2}\to O_{2}{}^{•-}$$
(4)$$O_{2}{}^{•-}+H^{+}\to {\mathrm{HO}}_{2}{}^{•}$$
(5)$${2\mathrm{HO}}_{2}{}^{•}\to H_{2}O_{2}+O_{2}$$
(6)$$e^{-}+H_{2}O_{2}\to {\mathrm{OH}}^{-}+{\mathrm{HO}}^{•}$$

OTC degradation efficiency in the visible light/TiO_2_ system at pH 8.6 was higher than that at pH 11.5 (Figure 1b), perhaps due to the better adsorption performance of OTC on TiO_2_ at pH 8.6, as shown in Figure 3. The point of zero charge (PZC) of the TiO_2_ used in this study is 6.9 [26]. As a result, the surface of TiO_2_ was positively charged at pH 2.0 and 5.7 but negatively charged at pH 8.6 and 11.5. OTC existed as protonated and deprotonated forms at pH 2.0 and 11.5, respectively, leading to electrostatic repulsion between OTC and TiO_2_ at these two pH values and consequently inhibiting OTC adsorption on TiO_2_. OTC adsorption at pH 8.6 was higher than that at pH 11.5, probably due to weaker electrostatic repulsion between TiO_2_ and OTC existing mainly as HOTC^−^ at pH 8.6 (Figure 3).

### 3.2. Identification of Reactive Oxygen Species

#### 3.2.1. Photolysis Process

The photolysis of OTC under visible light irradiation included two different pathways, direct photolytic degradation and indirect self-photosensitized degradation. To identify the ROS dominantly responsible for the self-sensitized OTC degradation during photolysis under the study’s visible light irradiation conditions, different ROS scavengers were added to the reaction solutions. As shown in Figure 4, NaN_3_, *p*-benzoquinone, catalase, and t-BuOH are respective scavengers of ^1^O_2_, O_2_^•−^, H_2_O_2_, and HO^•^. With the addition of *p*-benzoquinone, the OTC degradation was slightly inhibited at pH 8.6 but significantly depressed at pH 11.5. Visible light OTC decomposition in the presence of the other ROS scavengers was similar to OTC photolysis at both pH values. These results indicated that O_2_^•−^ was generated and that OTC photolysis mainly involved direct degradation and O_2_^•−^-induced self-sensitized degradation under the study’s visible light irradiation conditions. Niu et al. [17] came to the same conclusion.

#### 3.2.2. TiO_2_-Assisted Photosensitized Degradation Process

In OTC TiO_2_-assisted photodegradation under visible light, both direct photolysis and ROS-mediated oxidation contributed to OTC removal. To clarify the species of ROS and their contribution to OTC degradation, four different ROS scavengers were added to the visible light/TiO_2_ system at pH 8.6. The results are presented in Figure 5. OTC destruction was obviously inhibited by adding *p*-benzoquinone, as indicated by the presence of O_2_^•−^ and its reactivity with OTC, which were also demonstrated in OTC photolysis. With the individual addition of catalase and t-BuOH, the OTC removal was slightly depressed. This indicates the generation of H_2_O_2_ through O_2_^•−^ dismutation and HO^•^ from formed H_2_O_2_ dissociation, as shown in Equations (4)–(6). The presence of NaN_3_ hardly affected OTC degradation, revealing that the role of ^1^O_2_ in OTC decomposition could be negligible in the current study. Zhao et al. [12] observed similar results while confirming the OTC photocatalytic degradation mechanism with NF-TiO_2_ film under visible/solar light at pH 5.5 and 8.5. However, they did not investigate the contribution of O_2_^•−^ to OTC degradation. Based on our experiments, we could conclude that the ROS O_2_^•−^ was most responsible for indirect self-sensitized OTC degradation under the visible light/TiO_2_ system.

### 3.3. Effect of TiO_2_ Load on the Photosensitized Degradation of OTC

Determining the semiconductor dosage for photosensitized OTC degradation in the visible light/TiO_2_ system is important for two reasons. First, low TiO_2_ load may influence the removal efficiency of OTC. Second, high TiO_2_ load may interfere with the light absorption of OTC and increase the operation cost. To optimize the TiO_2_ load, OTC destruction was investigated using different TiO_2_ dosages (0.2, 0.5, 1.0, and 2.0 g L^−1^) under visible light at pH 5.7 and 8.6. As shown in Figure 6, OTC degradation increased with larger TiO_2_ doses at both pH values. The increased TiO_2_ load might improve OTC adsorption to its surface by exciting the state of OTC electrons and subsequently increasing the formation of ROS. An adverse effect of high TiO_2_ dosage was not observed in this study, indicating 2.0 g L^−1^ of TiO_2_ load was not excessive under the study’s conditions.

### 3.4. Mineralization of OTC

This study also investigated the mineralization of OTC in the visible light/TiO_2_ system at pH 8.6. The reaction solution’s change in TOC was used to reflect the mineralization efficiency of OTC. As shown in Figure 7, although OTC could be rapidly degraded, it could hardly be mineralized, and only approximately 4% TOC was removed after 5 h of visible light irradiation. This result indicates that in the presence of TiO_2_ under visible light, OTC may be transformed to intermediates but not CO_2_ and H_2_O in its photosensitized degradation process, as demonstrated in the following section. The previous works also found that OTC was difficult to mineralize in UV/H_2_O_2_ and UV/S_2_O_8_^2−^ systems after 10 h UV irradiation with approximately 9.5% and 15.7% TOC eliminated, respectively [27,28]. It can be concluded that OTC was not easily completely mineralized because of its complex, stable molecular structure. More work is needed to increase the mineralization efficiency of OTC.

### 3.5. Identification of Transformation Products and Reaction Pathways

Eight transformation OTC products were detected and identified in both photolysis and TiO_2_-assisted photosensitized degradation with visible light irradiation at pH 8.6. Their evolutions and reaction times are shown in Figure 8a,b. Although the speciation of these detected products was the same in both reaction systems, the apparent faster generation and further degradation of these products under visible light/TiO_2_ again indicates that radical reaction significantly contributes to OTC removal as discussed in Section 3.2.2. Given the transformation products identified, Figure 9 presents five different probable reaction pathways for photolytic and TiO_2_-assisted photosensitized OTC degradation under visible light: hydroxylation, quinonization, decarbonylation, demethylation, and dehydration. The same degradation paths were observed in OTC destruction by carbonate radicals (CO_3_^•−^) in the UV/H_2_O_2_/t-BuOH/Na_2_CO_3_ system, and all the products except *m*/*z* 493 were also detected [29].

In the photolysis and TiO_2_-assisted photosensitized OTC degradation under visible light at pH 8.6, direct degradation and O_2_^•−^-mediated self-sensitized degradation were the main reaction processes, as demonstrated above. Similar to the electrophilic nature of CO_3_^•−^, O_2_^•−^ might react readily with the electron-rich reactive sites in the OTC structure, such as the phenol moiety (ring D), through electron transfer, leading to the generation of hydroxylation product *m*/*z* 477 and quinonization product *m*/*z* 475 [29]. The formed *m*/*z* 477 could then be further hydroxylated to produce *m*/*z* 493, as shown in Figure 9. Decarbonylation could result from a loss of CO from the tricarbonyl system in ring A, resulting in the formation of the decarbonylation product *m*/*z* 433, which could then undergo hydroxylation to generate *m*/*z* 449 [27]. The generated *m*/*z* 449 might also be formed from *m*/*z* 477 via decarbonylation, or it could undergo dehydration, which might occur at C6-C5a, producing *m*/*z* 431 that could be further transformed into quinonization *m*/*z* 429 (Figure 9) [30]. Demethylation might occur in the dimethylammonium group at C4 in OTC direct photolysis under visible light, leading to the production of the demethylation product *m*/*z* 447. The generated *m*/*z* 447 was also detected in OTC photolysis under UV-254 nm irradiation [27]. Pereira et al. [21] found seven products in photocatalytic OTC degradation using TiO_2_ under simulated solar light irradiation. With the exception of *m*/*z* 477, all other products, including *m*/*z* 457, 441, 415, 386, 368, and 345, were not detected in our study, probably because of different reaction conditions and analysis methods. Jin et al. [31] detected four products from OTC photolysis under oxygen-free conditions with λ > 300 nm light irradiation; the current study identified the same products as Jin et al., with the exception of *m*/*z* 417.

## 4. Conclusions

This study investigated OTC degradation under visible light by photolysis and TiO_2_-assisted photosensitized and identified the predominant ROS and degradation mechanisms in these two reactions. We found that deprotonated OTC could be easily photolyzed under visible light and removal efficiency was enhanced with the addition of TiO_2._The enhancement in removal efficiency is likely due to TiO_2_ assisting self-photosensitized degradation, as verified by the replacement of TiO_2_ with the insulator SiO_2_ in the visible light/TiO_2_ system. According to these experiments, the ROS O_2_^•−^ contributed most to self-sensitized OTC degradation in both photolysis and TiO_2_-assisted photosensitized reactions under the study’s conditions. OTC destruction increased with TiO_2_ load in the visible light/TiO_2_ system at pH 5.7 and 8.6. The mineralization of OTC was limited in the TiO_2_ -assisted photosensitized degradation after 5 h of visible light irradiation despite its rapid decomposition. The LC-QTOF/MS detected eight reaction products in both photolysis and TiO_2_-assisted photosensitized OTC degradation under visible light. We subsequently proposed the five transformation pathways of hydroxylation, quinonization, decarbonylation, demethylation, and dehydration as OTC degradation mechanisms in these two reactions. This work suggests that visible light can induce OTC degradation with or without semiconductors under suitable conditions and provides a supplementary explanation for OTC removal in natural waters.

## Figures and Tables

**Figure 1 ijerph-19-15550-f001:**
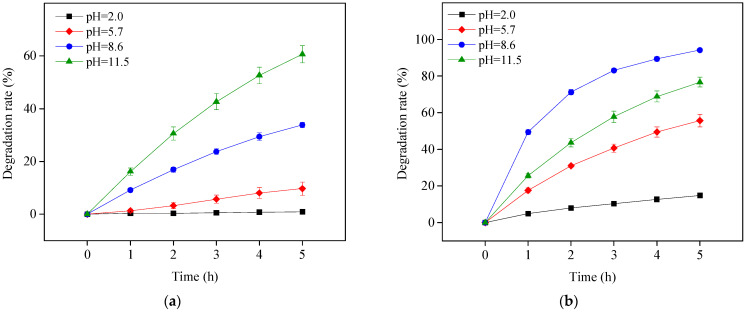
Photolysis (**a**) and TiO_2_-assisted photosensitized degradation (**b**) of OTC under visible light irradiation at different pH values. Experimental conditions: [OTC]_0_ = 10 mg L^−1^, [TiO_2_]_0_ = 0.5 g L^−1^, 50 mM phosphate buffer.

**Figure 2 ijerph-19-15550-f002:**
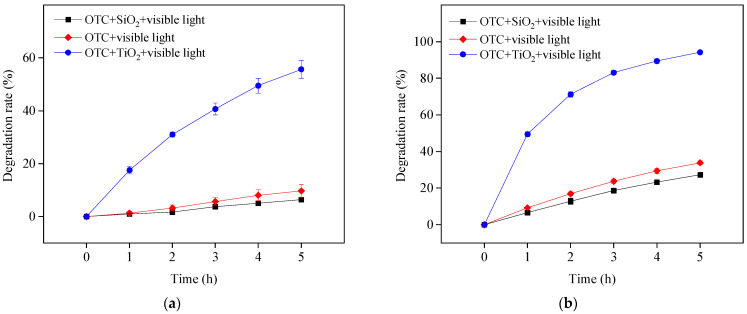
OTC degradation in the presence of SiO_2_ under visible light irradiation at pH 5.7 (**a**) and 8.6 (**b**). Experimental conditions: [OTC]_0_ = 10 mg L^−1^, [SiO_2_]_0_ = 0.5 g L^−1^, 50 mM phosphate buffer.

**Figure 3 ijerph-19-15550-f003:**
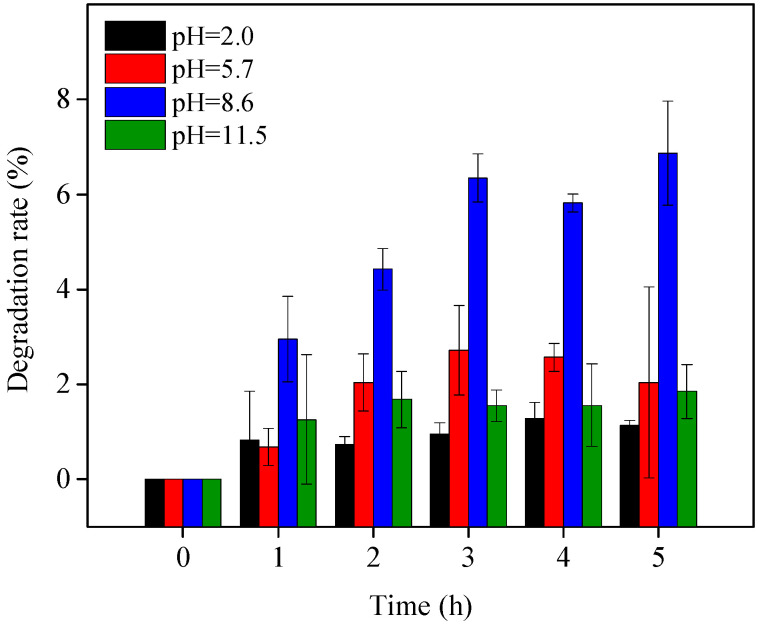
OTC adsorption on TiO_2_ at different pH values. Experimental conditions: [OTC]_0_ = 10 mg L^−1^, [TiO_2_]_0_ = 0.5 g L^−1^, 50 mM phosphate buffer.

**Figure 4 ijerph-19-15550-f004:**
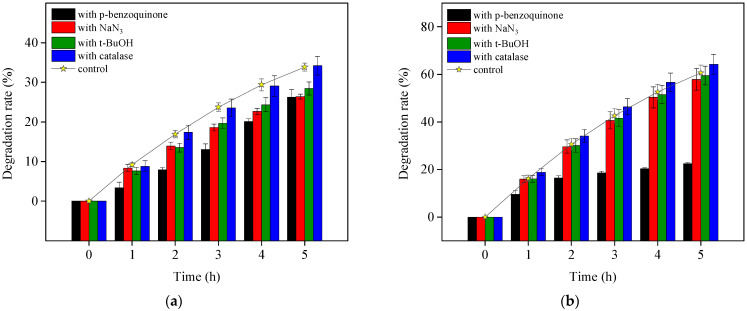
OTC photolysis with the addition of different ROS scavengers under visible light irradiation at pH 8.6 (**a**) and 11.5 (**b**). Experimental conditions: [OTC]_0_ = 10 mg L^−1^, [NaN_3_]_0_ = 10 mM, [t-BuOH]_0_ = 10 mM, [*p*-benzoquinone]_0_ = 5 mM, [catalase]_0_ = 12 unit L^−1^, 50 mM phosphate buffer.

**Figure 5 ijerph-19-15550-f005:**
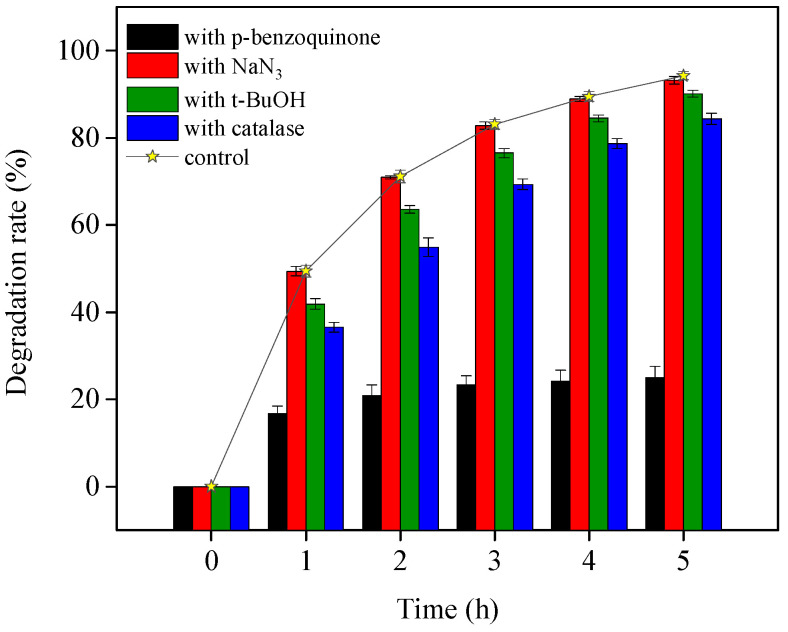
TiO_2_-assisted photosensitized OTC degradation with the addition of different ROS scavengers under visible light irradiation at pH 8.6. Experimental conditions: [OTC]_0_ = 10 mg L^−1^, [NaN_3_]_0_ = 5 mM, [t-BuOH]_0_ = 10 mM, [*p*-benzoquinone]_0_ = 5 mM, [catalase]_0_ = 12 unit L^−1^, 50 mM phosphate buffer.

**Figure 6 ijerph-19-15550-f006:**
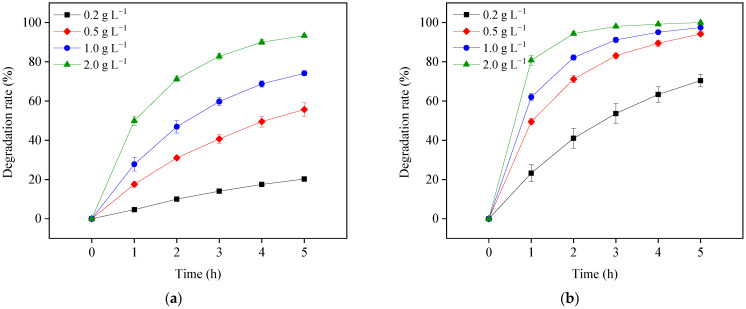
TiO_2_ load effects on OTC degradation in the visible light/TiO_2_ system at pH 5.7 (**a**) and 8.6 (**b**). Experimental conditions: [OTC]_0_ = 10 mg L^−1^, 50 mM phosphate buffer.

**Figure 7 ijerph-19-15550-f007:**
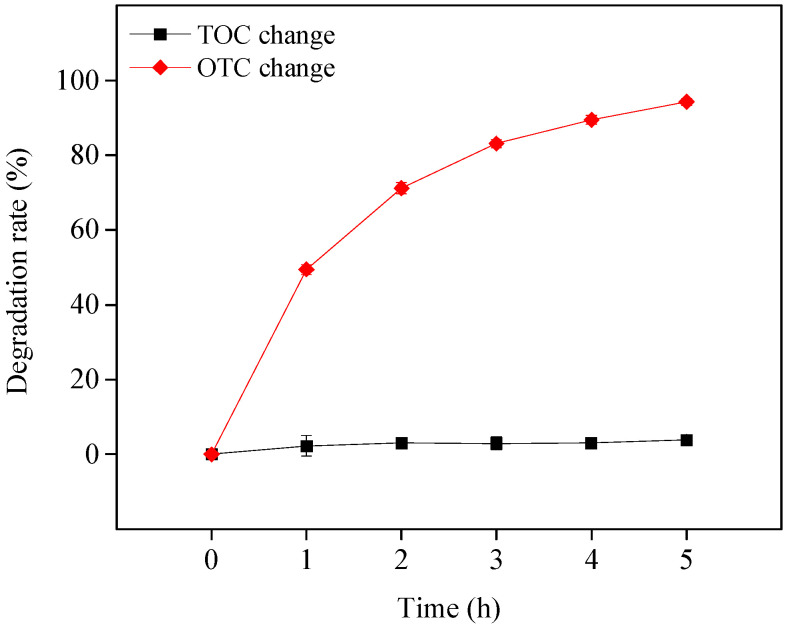
OTC mineralization in the visible light/TiO_2_ system at pH 8.6. Experimental conditions: [OTC]_0_ = 10 mg L^−1^, [TiO_2_]_0_ = 0.5 g L^−1^, 50 mM phosphate buffer.

**Figure 8 ijerph-19-15550-f008:**
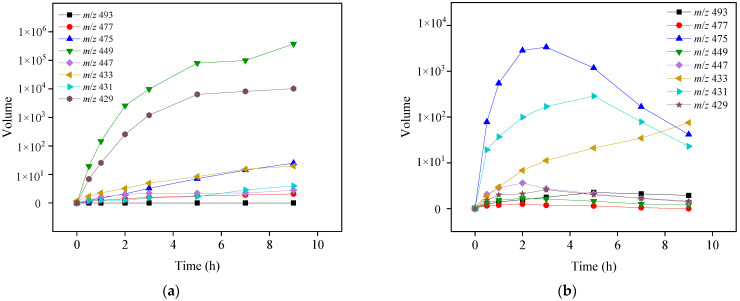
Evolutions of OTC transformation products by photolysis (**a**) and TiO_2_-assisted photosensitized degradation (**b**) under visible light irradiation at pH 8.6. Experimental conditions: [OTC]_0_ = 10 mg L^−1^, [TiO_2_]_0_ = 0.5 g L^−1^, 50 mM phosphate buffer.

**Figure 9 ijerph-19-15550-f009:**
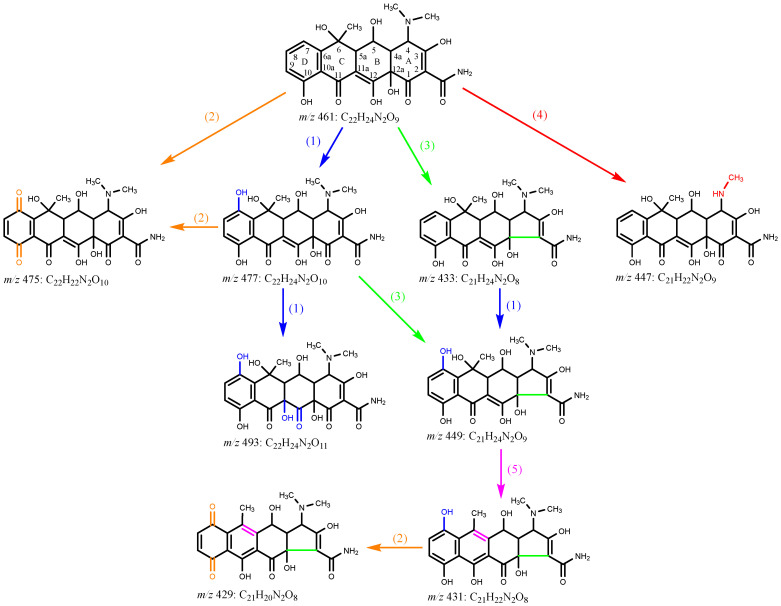
Probable photolytic and TiO_2_-assisted photosensitized degradation mechanisms of OTC under visible light irradiation at pH 8.6: (1) hydroxylation, (2) quinonization, (3) decarbonylation, (4) demethylation, and (5) dehydration. Experimental conditions: [OTC]_0_ = 10 mg L^−1^, [TiO_2_]_0_ = 0.5 g L^−1^, 50 mM phosphate buffer.

## Data Availability

The data are available upon reasonable request from the corresponding author.

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
