# Peer review of "Identification of Reactive Oxygen Species and Mechanism on Visible Light-Induced Photosensitized Degradation of Oxytetracycline"

_ijerph, 2022, doi:10.3390/ijerph192315550_

Round 1

Reviewer 1 Report

Yibo Zhang et al., investigated OTC degradation under visible light by photolysis and TiO2- assisted photosensitized and identified the predominant ROS and degradation mechanisms in these two reactions. They found that deprotonated OTC could be easily photolyzed under visible light and removal efficiency was enhanced with the addition of TiO2.

This paper has a very important contribution to the investigations trying to eliminate OTC from different aquatic environments and in that way reduce a potential drug-resistant risk of bacterial populations.

Studies to characterize and identify ROS in OTC are little explored and Yibo Zhang et al., addressed it in a good manner, proposing five transformation pathways of hydroxylation, quinonization, decarbonylation, demethylation and dehydration as OTC degradation mechanisms. They suggested that visible light can induce OTC degradation with or without semi-conductors under suitable conditions and provided a possible explanation for OTC removal in natural waters.

In general, this work was carefully performed, and this is an interesting contribution describing an alternative manner to remove OTC from natural waters.

I think this is a manuscript suitable for publication in the present form.

There are minor issues that are listed in order, as follow:

1) page 1, line 21-22: for a better understanding, rephrase this paragraph “we propose five potential reaction mechanisms for the photolytic and TiO2-assisted photosensitized degradation mechanisms of OTC under visible light were proposed, including hydroxylation, quinonization, decarbonylation, demethylation and dehydration.”

2) I suggest a figure to illustrate the workflow followed by the author so that it will be easier to understand.

3) in page 3, line 117-118: “For the degradation experiments, the irradiated solution (0.2 mL) was sampled after 0, 1, 2, 3, 4 and 5 h, mixed with 0.2 mL of methanol and filtered with a 0.2-μm syringeless filter before HPLC analysis” mentioned what the irradiated solution contains.

4) clarify how the solutions at different pHs were buffered. Was the buffer 50 mM phosphate buffer, just changing the pH? If so, is phosphate buffer suitable for such a huge range from 2.0 to 11.5?

5) Bring more clues or ideas to improve the mineralization of OTC since when it is degraded the percentage of mineralization is low as other reports.

Author Response

Thank you for taking the time to review our manuscript.

Reviewer 2 Report

The removal of oxytetracycline (OTC) by photolysis and photosensitized degradation using visible light was investigated, and the main reactive oxygen species (ROS) and reaction mechanism for OTC removal were also explored. The results are clearly shown and appropriately discussed. Some issues regarding this manuscript are listed below.

1.     The UV-Vis absorption spectra of OTC at different pHs studied should be presented.

2.     With respect to direct photolysis at different pHs, please see the work of Zhao et al 2013. In addition, other possible factors should be considered to justify the change in photolysis rate.

3.     The mechanism of ROS production under OTC photolysis should be clarified.

4.     From Figure 6 it can be concluded that the optimal TiO2 load was not reached and should be optimized.

5.     The authors should state if there is any advantage of using TiO2 in this system because no TOC removal was observed and the OTC degradation products seem to be similar.

Author Response

(The authors gave the same response as above.)

Round 2

Reviewer 2 Report

All the comments I addressed have been revised, and it can be accepted in current form.